# Brief Warm and Aldo-Keto Reductase Family *AspiAKR1B1* Contribute to Cold Adaptation of *Aleurocanthus spiniferus*

**DOI:** 10.3390/insects16010038

**Published:** 2025-01-02

**Authors:** Zhi-Fei Jia, Yan-Ge Cui, Meng-Yuan Liu, Jeremiah Joe Kabissa, Yong-Yu Xu, Zhi-Wei Kang, Zhen-Zhen Chen

**Affiliations:** 1State Key Laboratory of Wheat Improvement, College of Plant Protection, Shandong Agricultural University, Tai’an 271000, China; jiazf0525@163.com (Z.-F.J.); yxx936936@163.com (Y.-G.C.); lmysdau@163.com (M.-Y.L.); kabissaj@gmail.com (J.J.K.); xuyy@sdau.edu.cn (Y.-Y.X.); 2Tanzania Agricultural Research Institute (TARI), Mwanza 999132, Tanzania; 3College of Life Sciences, Hebei University, Baoding 071000, China

**Keywords:** *Aleurocanthus spiniferus*, *AspiAKR1B1*, fluctuating thermal regime, cold tolerance

## Abstract

The orange spiny whitefly seems to be spreading from low latitude to higher latitude areas. This research proved that brief warm pulses and *AspiAKR1B1* are key factors.

## 1. Introduction

Due to international trade and human activities, various invasive species of whiteflies (Hemiptera: Aleyrodidae) have become widespread, causing significant agricultural losses [1]. The orange spiny whitefly *Aleurocanthus spiniferus* (Quaintance) is recognized as an invasive pest originally from Southeast Asia [2]. This pest has been responsible for direct damage to over 90 plant species, including tea and grapes, leading to the occurrence of sooty blotch and creating major challenges for its management [3,4,5]. In just over a century, *A. spiniferus* has rapidly spread across Asia, Africa, the Americas, Australia, the Pacific Islands, and more, becoming a global pest [6,7,8,9]. In China, this pest was first identified in low latitude regions and is now increasingly found in higher latitudes, having established and overwintered in northern provinces like Shandong (34~38° N) and Shanxi (31~39° N) Province. Additionally, *A. spiniferus* has also been monitored as an invasive pest in Italy (36~47° N) and the Balkan Peninsula, with signs of northward spread towards France (42~51° N) [3,10]. This trend suggests a rapid increase in its adaptability to colder climates, raising concerns about further expansion into cooler regions.

If a species establishes in a temperate or colder climate zone, a key factor is the ability of enough individuals to survive in any cold periods that occur in those areas [11,12]. Research into cold tolerance is essential for evaluating the potential for population establishment and is a crucial part of the risk analysis for pest invasions [13,14,15]. Studies have shown that repeated cold stresses, characterized by exposing insects to short warm pulses during low-temperature stress (known as fluctuating thermal regime or FTR), can lead to better survival compared to continuous cold exposure [16]. It has been reported that *A. spiniferus* cannot survive below freezing [17]. Understanding how *A. spiniferus* survives and establishes a large population at low temperatures is crucial for preventing its spread to higher latitudes.

In autumn, changes in day length and in temperature help insects to anticipate the arrival of winter, allowing them to take defensive measures in advance. Biochemical reactions, changes in cell function, and the expression of cold-resistant genes in insects can rapidly enhance their cold tolerance in a short time frame [18]. Low molecular weight substances such as trehalose, sorbitol, inositol, and fructose play an essential role in cold tolerance by enhancing the stability of cell membranes and proteins, thereby preventing osmotic damage to cells [19]. Fat is one of the main nutrients of insects, and its function is to provide energy. Glycerol, a metabolite of fat, is the most common polyol in overwintering insects or in insect responses to cold conditioning [20]. It can lower the supercooling point (SCP) of insects and enhance their cold tolerance [21].

The aldo-keto reductase (AKR) superfamily comprises a large group of proteins, found in all kingdoms of life [22]. Aldehyde reductase (AKR1A) reduces 3-deoxyglucosone and methylglyoxal, which are active intermediates involved in glycation, a non-enzymatic glycosylation reaction. AKR1A also plays a role in the denitrosylation of S-nitrosylated glutathione and coenzyme A [23,24]. Aldose reductase (AKR1B) is involved in the conversion of glucose to sorbitol through the polyol pathway [25]. Polyols and sugars are the most prevalent cryoprotectant molecules in overwintering insects [26], but the physiological functions of AKR1A and AKR1B in insect cold tolerance are currently not fully understood.

This study examines potential reasons for the improved cold tolerance of *A. spiniferus* by assessing the effects of brief warm pulses on their survival rate at −7 °C, analyzing seasonal patterns of cold-resistant substances and genes, and basic functions of the aldose reductase gene *AspiAKR1B1*. Preventing the further spread of *A. spiniferus* to higher latitudes is theoretically supported by this research.

## 2. Materials and Methods

### 2.1. Insect Collection

*A. spiniferus* was collected from the tea plant in plantations managed by Shandong Qianrun Ecological Agriculture Development Co., Ltd. in Tai’an, Shandong Province, China (32°08′ N, 117°43′ E).

### 2.2. Supercooling (SCP) and Freezing Point (FP)

Nymphs (3rd and 4th instar) and pseudopupae of *A. spiniferus* were collected in May, July, September, and during overwintering (November, January, March) to determine the seasonal differences in cold tolerance. Nymphs (3rd and 4th instar), pseudopupae, and adults (both female and male) were collected in November to examine cold tolerance across different developmental stages. The thermocouple method was used to measure the supercooling point (SCP) and freezing point (FP), with 30 individuals tested each time, and each individual was measured only once. The specific operation of the SCP and FP measurement is as follows: a small amount of petrolatum jelly is used as an adhesive to fix the test insect on the thermocouple probe, each probe fixes a single test insect at a time, and then places it in a variable temperature bath. The temperature in the bath is transmitted to the connected computer through the sensor, recording the temperature change at each moment and capturing the appearance of SCP and FP.

### 2.3. Low-Temperature Survival Rate

Nymphs (1st, 2nd, 3rd, and 4th instar), pseudopupae, and adults were collected from the field in July and placed in a refrigerator at −7 °C (simulating the minimum temperature of suitable areas of *A. spiniferus* in China. Data source: https://data.stats.gov.cn/index.htm (accessed on 13 January 2023) for 12, 24, 36, 48, 60, and 72 h. Subsequently, the tested insects were placed in an incubator for 24 h (25 °C, 70% RH, and 16L:8D). The number of surviving individuals was recorded, and the survival rate along with the median lethal time were calculated. Each treatment included 3 biological replicates, with 30 individuals per replicate. The LT50 was derived from the mortality rate corrected according to Abbott’s formula.

### 2.4. Brief Warm Pulses Test

First and second instar nymphs and overwintering nymphs (third and four instar) were subjected to brief warm pulse treatments in environmental incubators (as detailed in Table 1). Following this, they were placed in environmental incubators at 25 °C with 70% RH and a 16L:8D photoperiod for 24 h to assess the survival rate. Each treatment included 3 biological replicates, with 30 individuals for each replicate.

### 2.5. Moisture Content

Fourth instar nymphs of *A. spiniferus* were collected in May, July, September, and during overwintering (November, January, March) in the field. The fresh mass (FM) was first accurately weighed on an electronic balance. Subsequently, the nymphs were dried in a constant temperature drying oven at 60 °C for 48 h, after which the dry mass (DM) was recorded. Each treatment included 3 biological replicates, with each replicate containing 100 tested insects. Moisture content = (FM − DM)/FM × 100%.

### 2.6. Fat Content

A fat content test was conducted following the method outlined by Colinet et al. [27]. Fourth instar nymphs were collected in May, July, September, and during overwintering (November, January, March). After being dried at 60 °C for 48 h, their dry weight (DW) was measured. The dried tissues were placed in a 2 mL centrifuge tube, and 1.5 mL of chloroform and methanol (chloroform:methanol = 2:1) was added. The mixture was ground using a grinding cup and centrifuged at 8000× *g* for 10 min at room temperature. The supernatant was discarded, and another 1.5 mL mixture of chloroform and methanol was added to the precipitate, followed by another centrifugation. The remaining precipitate was dried at 60 °C for 48 h to determine the lean dry mass (LDW). Each treatment included 3 biological replicates, each with 100 tested insects. The fat content (FC) was calculated as follows: FC = (DW − LDW)/DW × 100%.

### 2.7. Polyol and Sugar Content

The determination of polyol and sugar content was primarily based on the methods of Zhang et al. [28]. Fourth instar nymphs were collected in November and March. They were placed in a 1.5 mL centrifuge tube, and 0.4 mL of pre-cooled mixed solution (methanol:ethanol:chloroform = 8:1:1) was added. After thorough grinding, 2 μL of internal standard decanoic acid was included. Following treatment in a refrigerator at 20 °C for 1 h, the mixture was centrifuged at 8000× *g* for 10 min at room temperature. The supernatant was transferred to a new tube, and 40 μL oxymethyl hydrochloride amine salt pyridine solution was added after spin drying. The mixture was incubated at 37 °C for 1 h. Then, 50 μL of N-methyl-N-(trimethylsilyl) trifluoroacetamide (MSTFA) was added and the mixture was shaken in a temperature-controlled environment at 37 °C for 30 min. Finally, 0.4 mL of n-hexane was added, and it was centrifuged at 14,000× *g* for 10 min at room temperature. The supernatant was analyzed using a gas chromatography–flame ionization detector (GC-FID), and the GC was an Agilent 7890B model. Each treatment included 3 biological replicates, with 100 test insects per replicate. The concentration ranges for GC–MS standards were the following: 0.1–30 μg/vial for glycerol, 0.1–20 μg/vial for trehalose, and 0.1–2 μg/vial for glucose, sorbitol, and erthrose.

### 2.8. Expression Levels of Cold-Resistant Genes

Fourth instar nymphs were collected in September and during overwintering (November, January, March). All gene sequences and expression data were obtained from the NCBI Short Read Archive (SRA) database, with the accession number being PRJNA792195. Primers are listed in Appendix A. Ribosomal protein S28 (RPS28) was used to normalize the target gene expression and to correct for sample-to-sample variation [5]. The total RNA was extracted using the Total RNA Extraction Kit (Vazyme, Nanjing, China), and cDNA was obtained using the cDNA synthesis kit (Vazyme, Nanjing, China). The relative transcript abundance was examined using the CFX96 Real-Time System (Bio Red, Hercules, CA, USA). The reaction system consisted of 20 μL:10 μL of 2×ChamQ SYBR qPCR Master Mix (Vazyme, Nanjing, China), 1 μL of each primer, and 8 μL of cDNA (10-fold dilution). A standard three-step PCR method was employed. Stage 1 was initial denaturation at 94 °C for 30 s, 1 cycle. Stage 2 was 40 cycles at 95 °C for 5 s and at 60 °C for 30 s. The melting curve of PCR products was detected at 95 °C for 15 s, 60 °C for 1 min, and 95 °C for 15 s. Each treatment included three biological replicates and three technical replicates. The mRNA relative expression levels were calculated using the comparative 2^−ΔΔCt^ method [29].

### 2.9. RNAi of AspiAKR1B1

The TAKARA pMDTM18-T Vector Cloning Kit (Japan) was used for DNA fragments. The recombinant plasmid was transformed into DH5*α* competent cells (Vazyme, Nanjing, China) and cultured on the Luria-Bertani (LB) plate (containing 100 μg mL^−1^ kanamycin) at 37 °C for 16 h. White monoclonal bacterial colonies were selected for bacterial colony PCR and sequencing. The correct recombinant plasmids were extracted using the TIANGEN prep Mini Plasmid Kit (Beijing, China). Specific primers (listed in Appendix A) featuring T7 promoter sequences were designed to synthesize double-stranded RNA (dsRNA) of *AspiAKR1B1* using the T7 RiboMAX™ Express RNAi kit (Promega, Madison, WI, USA), according to the manufacturer’s instructions.

Thirty adult *A. spiniferus* that had been starved for 1 h were transferred to a food-grade plastic cup (38 mm in top diameter, 30 mm in bottom diameter, and 30 mm high). The sealing film (Bemis, Inc., Neenah, WI, USA) was stretched as thin as possible and used to seal the cup. The dsRNA was diluted to 100 ng μL^−1^ with 30% sucrose solution, and 20 μL of dsRNA solution was added dropwise to each sealing film, which was then sealed again using a new sealing film [30]. The sealing film allows the insect to feed from the cup. After 24 h in environmental incubators at 25 °C, the tested insects were collected for the determination of survival, fat content, mortality, and gene expression. Feeding *dslta* (Mus musculus lymphotoxin A, GenBank: XM_006536550.2) was used as the control. Each treatment was replicated three times.

### 2.10. Data Analysis

All data were analyzed with GraphPad Prism 9 software (GraphPad Software, San Diego, CA, USA). Data on survival, fat content, gene expression levels, and moisture content were analyzed by one-way analysis of variance (ANOVA). Means were compared using Tukey’s-b multiple range test. Data on polyol and sugar content as well as RNA interference (RNAi) were analyzed using two-tailed student’s *t* test.

## 3. Results

### 3.1. The Low-Temperature Survival Ability of A. spiniferus at −7 °C

The survival rate of *A. spiniferus* decreased with prolonged exposure to low temperatures. When exposed to −7 °C for 12 h, the survival rate of second instar nymphs was the highest at 81.67 ± 4.41%, while first instar nymphs had the lowest survival rate at 70 ± 5%. After 24 h at −7 °C, the survival rate of third instar nymphs was the highest at 66.67 ± 4.41%, and that of first instar nymphs was the lowest at 56.67 ± 3.33%. After 36 h of low temperature, the survival rate of second instar nymphs was the highest at 60 ± 5.77%, while that of third instar nymphs was the lowest at 45 ± 8.66%. Following treatments for 48, 60, and 72 h, the survival rates of fourth instar nymphs were the highest at 51.67 ± 4.41%, 35 ± 5.77%, and 20 ± 2.89%, respectively. In contrast, the first instar nymphs exhibited the lowest survival rates at 34.33 ± 5.81%, 21.33 ± 1.33%, and 5 ± 0%, respectively (Figure 1). The median lethal time for the fourth instar nymph was the longest at 44.054 h, while the median lethal time for adults was the shortest at 2.813 h (Table 2).

### 3.2. The Brief Warm Pulses Improved the Survival Rate of A. spiniferus

The three different brief warm pulse treatments had no significant impact on the survival rates of first and second instar nymphs (Figure 2a,b). The brief warm pulses of 25 °C (T2) during low-temperature stress at −7 °C significantly improved the survival rate of the third instar nymph (*t* = 3.471, *df* = 4, *p* = 0.05; CK was 15%, T2 was 28.67%) and fourth instar nymph (*t* = 5.418, *df* = 4, *p* = 0.004; CK was 20%, T2 was 45.67%) (Figure 2c,d). While the survival rate of the T1 and T3 treatments also increased, they did not achieve statistical significance level.

### 3.3. Seasonal Differences in Cold Tolerance of A. spiniferus

Significant differences were observed in the supercooling point (SCP) (*F* = 50.77, *df* = 4, *p* < 0.001) and freezing point (FP) (*F* = 59.79, *df* = 4, *p* < 0.001) among the different developmental stages of *A. spiniferus* natural populations in November (Figure 3a,b). The SCP and FP of adult *A. spiniferus* (female and male) were significantly higher than those of third and fourth instar nymphs, as well as pseudopupae (*p* < 0.05). No significant differences were found between third and fourth instar nymphs and pseudopupae (*p* > 0.05).

In November, the SCP and FP of the overwintering nymphs (third and fourth instar) and pseudopupae were at their lowest levels. The SCP of the third instar nymph in November (−27.04 °C) was significantly lower than in May (−25.29 °C; *t* = 4.138, *df* = 23, *p* = 0.006) and July (−24.65 °C; *t* = 3.336, *df* = 23, *p* = 0.04) (Figure 3c). Similarly, the FP was significantly lower in November (−26.71 °C) than in May (−24.98 °C; *t* = 4.454, *df* = 23, *p* = 0.003) (Figure 3f). For the fourth instar nymph, the SCP in November (−27.02 °C) was significantly lower than in July (−23.14 °C; *t* = 5.570, *df* = 17, *p* < 0.001) (Figure 3d), and the FP (−27.38 °C in November) was significantly lower than in March (−24.63 °C; *t* = 4.558, *df* = 17, *p* = 0.004), May (−24.02 °C; *t* = 5.995, *df* = 17, *p* < 0.001), and July (−21.53 °C; *t* = 8.810, *df* = 17, *p* < 0.001) (Figure 3g). The SCP of pseudopupae in November (−26.63 °C) was also significantly lower than in July (−22.37 °C; *t* = 4.972, *df* = 17, *p* < 0.001) (Figure 3e), and the FP (−26.12 °C in November) was significantly lower than that in March (−23.57 °C; *t* = 3.427, *df* = 17, *p* = 0.05), May (−23.21 °C; *t* = 3.387, *df* = 17, *p* = 0.05), and July (−21.17 °C; *t* = 6.833, *df* = 17, *p* < 0.001) (Figure 3h).

### 3.4. Seasonal Differences in Substance Contents of A. spiniferus

The moisture content was the lowest in November and January, while it was the highest in March (Figure 4a). The fat content of *A. spiniferus* nymphs peaked in November and January, but was the lowest in July (Figure 4b). Glycerol content was significantly higher in November than in March (*t* = 6.577, *df* = 4, *p* = 0.01); conversely, levels of glucose, sorbitol, and trehalose were significantly lower in November than in March. There was no significant difference in erythrose content between the two months (Figure 5).

### 3.5. The Function of AspiAKR1

The expression level of *AspiAKR1A3* was significantly higher in November, January, and March compared to September (Figure 6c). Meanwhile, *AspiAKR1B1* showed the highest expression in November and January, with the lowest levels observed in March (Figure 6d). There were no significant seasonal variations in the expression patterns of *AspiAKR1A1*, *AspiAKR1A2*, *AspiAKR1B2*, and *AspiAKR1B3* (Figure 6a,b,e,f). The knockdown of *AspiAKR1B1* resulted in a significant reduction of 37.139%, compared with the *dslta* treatment (Figure 7a), and significantly increased the SCP (*t* = 4.639, *df* = 38, *p* < 0.001) and FP (*t* = 4.395, *df* = 38, *p* < 0.001) of adults (Figure 7b,c). Meanwhile, the knockdown of *AspiAKR1B1* significantly increased the mortality of adults at −7 °C for 30 min (*F* = 0.712, *t* = 4.811, *df* = 4, *p* = 0.009) and 60 min (*F* = 1.146, *t* = 4.979, *df* = 4, *p* = 0.008) (Figure 7d).

## 4. Discussion

The primary challenge for insects experiencing cold stress is to survive and subsequently resume growth, development, and reproduction when the conditions become more favorable [31]. *A. spiniferus* has established in northern provinces like the Shandong and Shanxi Provinces of China. The average minimum temperature of the coldest month in the same latitude provinces is −5~−9 °C by consulting the China Statistical Yearbook [https://data.stats.gov.cn/index.htm (accessed on 13 January 2023)]. Therefore, this research takes −7 °C as the model temperature. The survival rate of *A. spiniferus* decreased with prolonged exposure to −7 °C, with the mortality rate exceeding 80% after three days (Figure 1). Intermittent exposure to a temperature of 5, 15, and 25 °C significantly enhanced the survival rate of overwintering nymphs (third and fourth instar nymphs) at −7 °C (Figure 2). These findings align with previous studies that demonstrated the beneficial effects of warm periods found in other species [16,27,32,33,34]. Notably, at the highest pulse (25 °C), the survival rate of overwintering nymphs was significantly higher than the control group. Although the gradually increasing temperatures (T1, T2) could also improve the survival rate, the effects were not statistically significant. These warm periods that interrupt an extended cold spell may contribute significantly to the ability of *A. spiniferus* to adapt to lower temperatures and gradually expand its range further north.

In this study, the SCP and FP of third and fourth instar nymphs, and the pseudopupae of *A. spiniferus* were the lowest in November, and significantly lower than adults (Figure 3). Additional measurements have shown that the cold tolerance of the invasive pest *Aleurodicus dispersus* (Homoptera: Aleyrodidae) nymph is higher than that of adults, increasing the risk of their northward invasion [35]. The *A. spiniferus* is a freeze-avoiding species, which means that it can survive at low temperatures above its SCP by lowering the FP of its extracellular fluid [31]. Thus, SCP is a critical metric for assessing cold tolerance.

Cryoprotective dehydration is a strategy that many overwintering insects employ to enhance survival at low temperatures [36]. As *A. spiniferus* nymphs enter the overwintering phase, their moisture content gradually decreases, while it increases again after overwintering. This suggests that the overwintering nymph may mitigate low temperature damage by reducing their internal moisture. Conversely, fat content increased during overwintering and declined thereafter. In winter, these insects can regulate chemicals associated with cold tolerance (moisture, fat, carbohydrates, etc.) through physiological and biochemical reactions, and convert sugars and other chemicals into fat [37]. Fatty compounds can also be hydrolyzed into antifreeze agents such as glycerol to enhance cold tolerance, as observed in *Dendroctonus armandi* [38], *Megabruchidius dorsalis* [39], and *Anoplophora glabripennis* [40]. Therefore, overwintering nymphs of *A. spiniferus* may protect themselves by increasing the fat content to adapt to cold environments.

Glycerol is a critical cryoprotective agent for insects [41]. The glycerol content in *A. spiniferus* nymphs during overwintering (November) was significantly higher than at the end of overwintering (March) (Figure 5), indicating that it may be an essential cold resistant substance for *A. spiniferus* nymphs. Many insects accumulate glycerol during overwintering to safeguard their bodies and to minimize frost damage. For instance, the glycerol content of *Chilo suppressalis* (Lepidoptera: Pyralidae) larvae increases significantly during overwintering, enhancing their cold tolerance [42]. The accumulation of glycerol in overwintering eggs of *Gomphocerus sibiricus* (Orthoptera: Acrididae) not only helps to withstand a cold environment but also provides reserve energy for subsequent development [43].

Polyhydric alcohols, such as arabitol, erythritol, sorbitol, mannitol, along with sugars like trehalose and glucose have often been commonly associated with insect cold tolerance [31]. The mobilization of glucose responds to cold stress and is linked to improved insect cold tolerance [44]. Elevated glucose levels in response to cold stress have been documented in *Drosophila melanogaster* [45], *Sarcophaga crassipalpis* [46], and other insects [47,48], with a correlation to enhanced cold tolerance. In a study of the beetle *Alphitobius diaperinus,* changes in the concentration of glucose were observed only after the cold exposure period ended and returned to baseline levels after the subsequent warm period [48]. This finding is consistent with our observation that, at the end of overwintering (March), the levels of glucose, sorbitol, and trehalose significantly increased (Figure 5). However, due to the complex nature of metabolic pathways and processes, it is challenging to determine whether these changes are beneficial (protective) or harmful (dysregulated) signals [49].

The seasonal expression profile of cold-resistant genes showed that the expression level of *AspiAKR1A3* was significantly higher in November, January, and March than in September (Figure 6). AKR1A3 plays a critical role in the response to environmental stress across multiple species. It functions as an oxidoreductase in the liver and kidneys of mammals (mice) [50]; increased lipid peroxidation in frog tissues is accompanied by an increase in AKR1A3 levels [51]. In African clawed frog (*Xenopus laevis*), dehydration induces organ-specific increases in AKR1A3 levels [52]. Although the role of AKR1A3 in insects is seldom explored, it is speculated that it may be involved in the oxidative stress response of *A. spiniferus* under low-temperature stress.

The expression level of *AspiAKR1B1* was significantly higher during overwintering (November and January) than at the end of overwintering (March). When *AspiAKR1B1* was knocked down, the SCP and FP of *A. spiniferus* increased, and the mortality increased after 30 and 60 min at −7 °C (Figure 7), indicating that AKR1B1 may play a positive role in coping with low-temperature stress. In mammals, AKR1B1 is involved in the polyol pathway and catalyzes the reduction of glucose to sorbitol [25]. In this study, the up-regulation of *AspiAKR1B1* coincided with a decrease in glucose. AKR1B1 is vital for glucose metabolism and it also aids in reducing superoxide and toxic substances [53]. Thus, it is speculated that AKR1B1 may help in maintaining membrane potential balance under low-temperature stress and act as a crucial component of the antioxidant and detoxification system that mitigates oxidative damage in such conditions. Furthermore, glyceraldehyde could serve as a substitute substrate for AKR1B1 [54], suggesting that *AspiAKR1B1* may also be a key gene in glycerol synthesis or metabolism pathways.

## 5. Conclusions

This study found that brief warm pulses could improve the survival rate of overwintering nymphs of *A. spiniferus* with the highest cold tolerance from September to November. Fat and glycerol might be directly related to the enhanced cold tolerance of *A. spiniferus*. This research provides the first evidence that AKR1B1 is involved in the cold tolerance of insects, although further investigation is needed to clarify its specific functions in the insects’ response to low-temperature stress.

## Figures and Tables

**Figure 1 insects-16-00038-f001:**
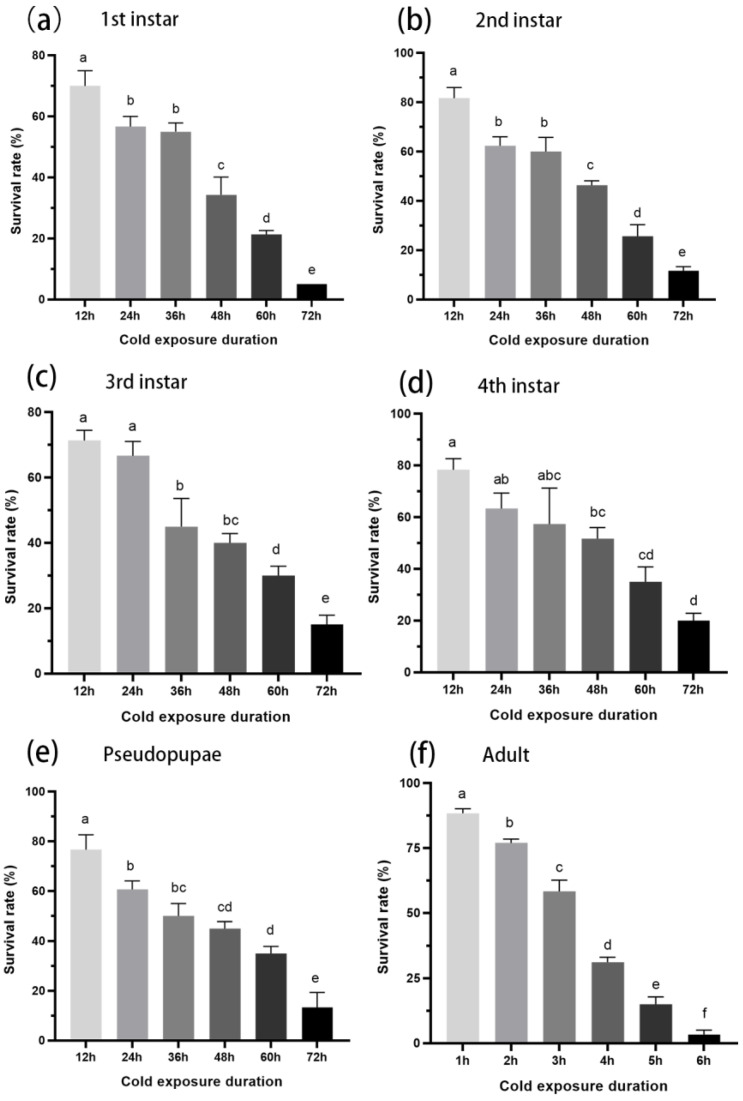
Survival rate of *Aleurocanthus spiniferus* at −7 °C. (**a**): 1st instar nymph, (**b**): 2nd instar nymph, (**c**): 3rd instar nymph, (**d**): 4th instar nymph, (**e**): pseudopupae, (**f**): adults. Data in the figure are means ± SE. Different letters at the top of columns mean significant differences (*p* < 0.05) detected by one-way analysis of variance, means were compared using Tukey’s-b multiple range test.

**Figure 2 insects-16-00038-f002:**
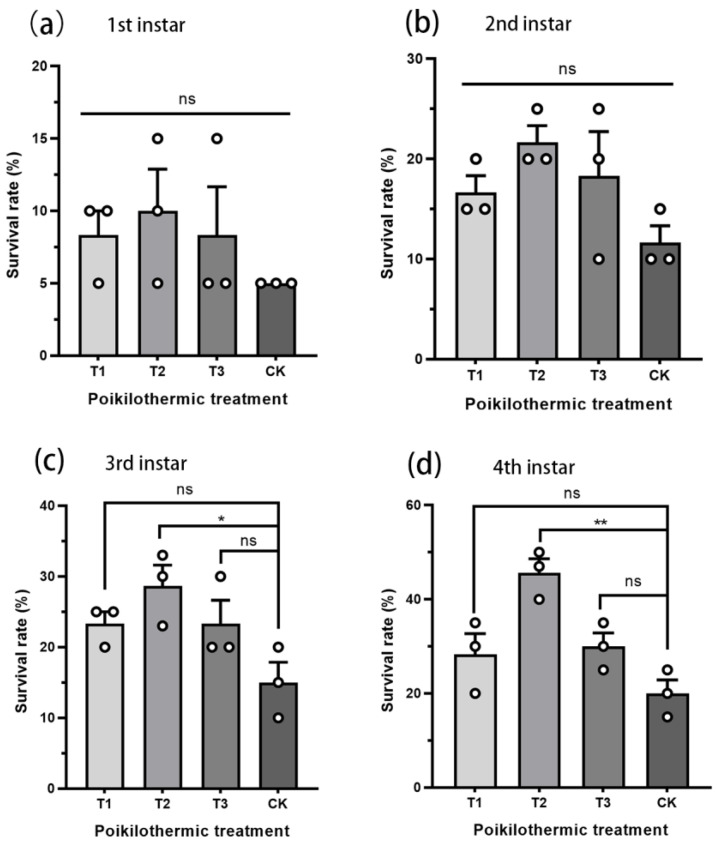
Survival rate of *Aleurocanthus spiniferus* at different brief warm pulse treatments. (**a**): 1st instar nymph, (**b**) 2nd instar nymph, (**c**) 3rd instar nymph, (**d**) 4th instar nymph. The asterisk represents significant differences between treatments using one-way analysis of variance, means were compared using Tukey’s-b multiple range test, * *p* < 0.05, ** *p* < 0.01, and ns means no significant difference (*p* > 0.05).

**Figure 3 insects-16-00038-f003:**
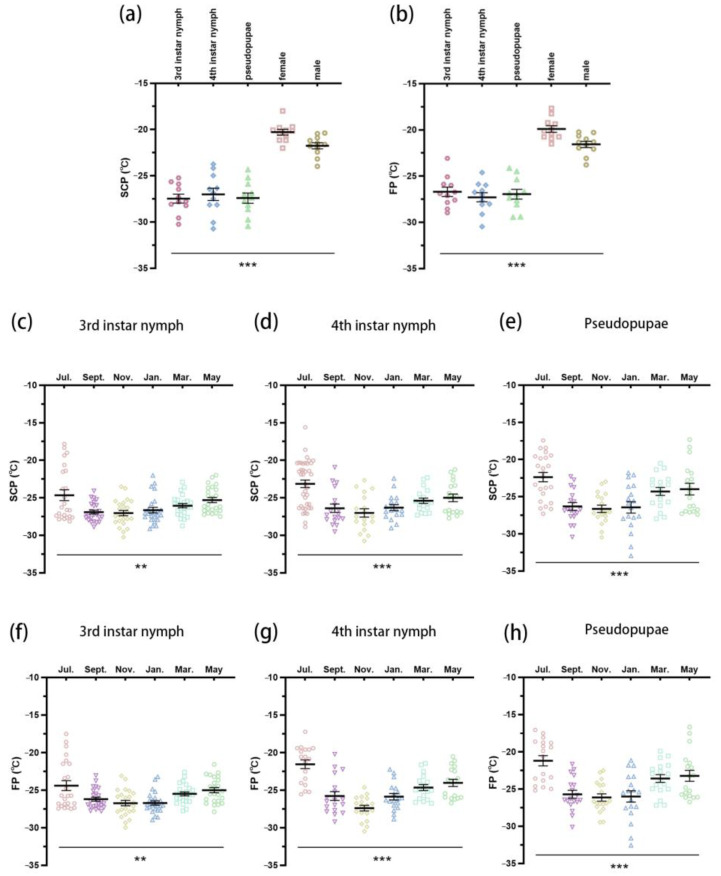
The SCP and FP of *Aleurocanthus spiniferus*. The SCP (**a**) and FP (**b**) of 3rd instar nymph, 4th instar nymph, pseudopupae, female and male from the natural population of *Aleurocanthus spiniferus* in November. The SCP and FP of 3rd instar nymph (**c**,**f**), 4th instar nymph (**d**,**g**), pseudopupae (**e**,**h**) from the natural population of *Aleurocanthus spiniferus* in different months. Different shapes and colours represent different developmental stages (**a**,**b**) and months (**c**–**h**). The asterisk represents significant differences between treatments using one-way analysis of variance, means were compared using Tukey’s-b multiple range test, ** *p* < 0.01, *** *p* < 0.001.

**Figure 4 insects-16-00038-f004:**
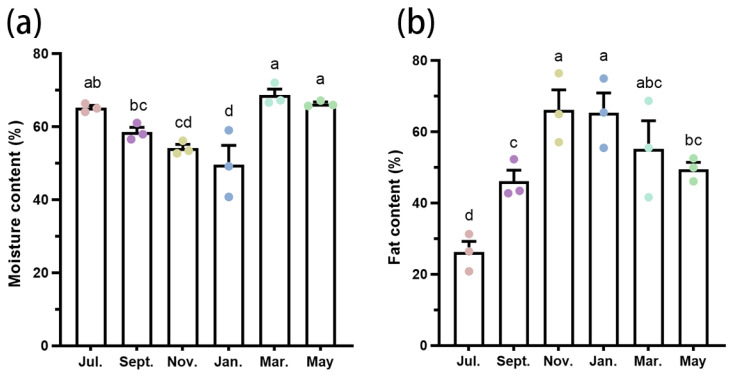
Moisture content (**a**) and fat content (**b**) of *Aleurocanthus spiniferus* nymph in different months. Data in the figures were means ± SE. Different colored circles represent different months. The different letters at the top of columns mean significant difference (*p* < 0.05) detected by one-way analysis of variance, means were compared using Tukey’s-b multiple range test.

**Figure 5 insects-16-00038-f005:**
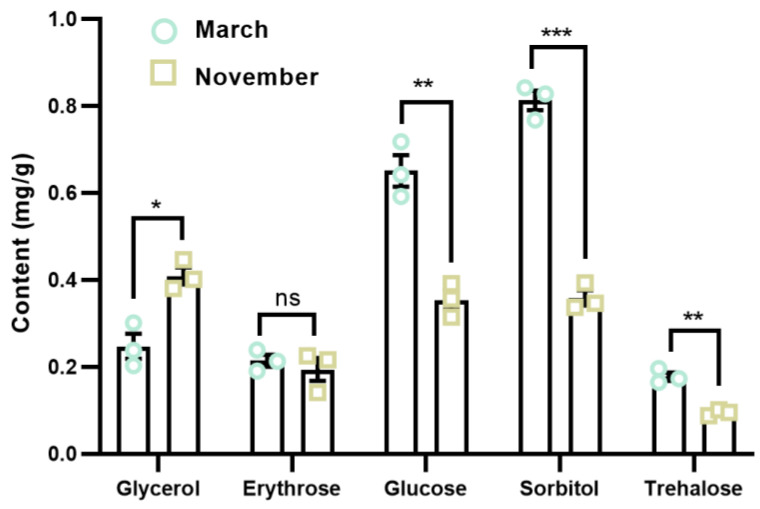
The polyol and sugar content of *Aleurocanthus spiniferus* nymph in March and November. The asterisk represents significant differences of low molecule substance contents between March and November using two-tailed Student’s *t*-test, * *p* < 0.05, ** *p* < 0.01, *** *p* < 0.001, while ns means no significant difference (*p* > 0.05).

**Figure 6 insects-16-00038-f006:**
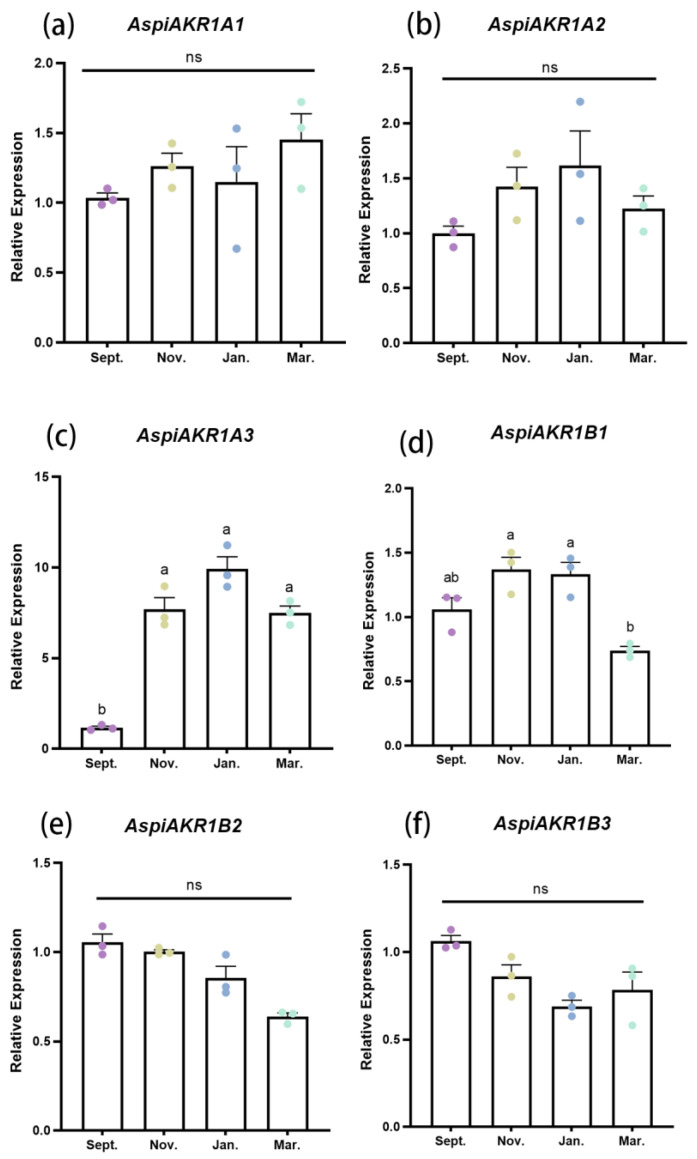
Expression levels of *AspiAKR1A* and *AspiAKR1B* in different months. (**a**): *AspiAKR1A1*, (**b**): *AspiAKR1A2*, (**c**): *AspiAKR1A3*, (**d**): *AspiAKR1B1*, (**e**): *AspiAKR1B2*, (**f**): *AspiAKR1B3.* Data in the figure are means ± SE. Different colored circles represent different months. The different letters at the top of columns indicate significant difference of *AspiAKR1* relative expressions between different months (*p* < 0.05) detected by one-way analysis of variance, means were compared using Tukey’s-b multiple range test. ns means no significant difference (*p* > 0.05).

**Figure 7 insects-16-00038-f007:**
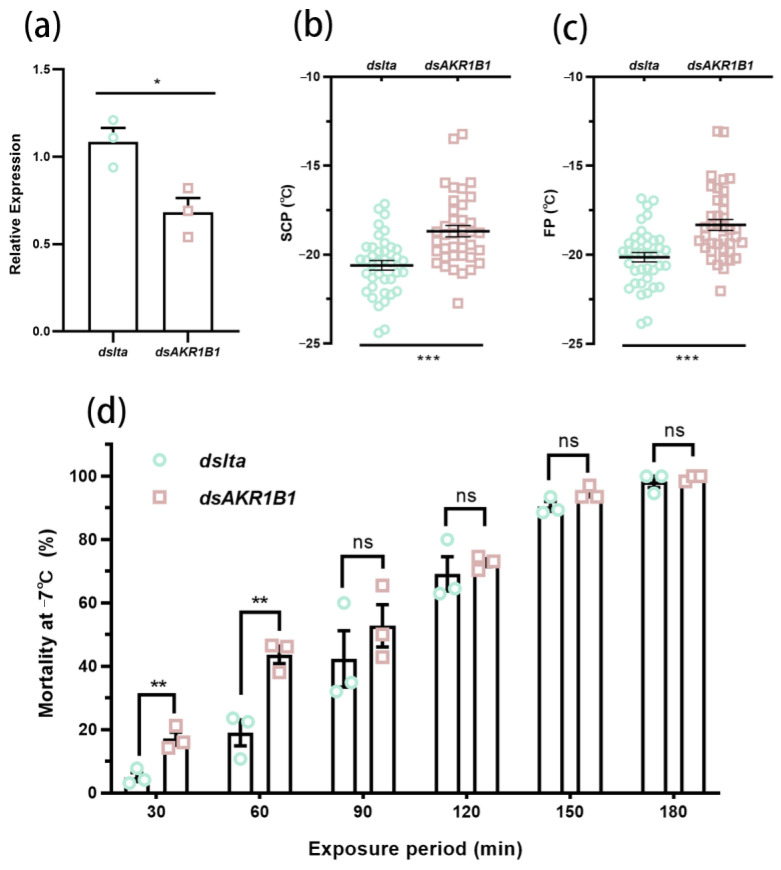
RNAi treatment of *AspiAKR1B1*. (**a**): Effect of RNAi treatment on the transcript levels of *AspiAKR1B1* and *lta*. (**b**,**c**): The SCP and FP of *dsAKR1B1* and *dslta* treatments. (**d**): The mortality of *dsAKR1B1* and *dslta* treatments at −7 °C. Asterisks indicate significant differences between treatments using two-tailed Student’s *t*-test, * *p* < 0.05, ** *p* < 0.01, *** *p* < 0.001, while ns means no significant difference (*p* > 0.05).

**Table 1 insects-16-00038-t001:** Brief warm pulse treatments.

Treatment	Temperature/Duration					
T1	−7 °C/22 h	5 °C/2 h	−7 °C/22 h	15 °C/2 h	−7 °C/22 h	25 °C/2 h
T2	−7 °C/23 h	25 °C/1 h	−7 °C/22 h	25 °C/2 h	−7 °C/20 h	25 °C/4 h
T3	−7 °C/ 22 h	15 °C/1 h	25 °C/1 h	(Repeat the above three times)		
CK	−7 °C/72 h					

**Table 2 insects-16-00038-t002:** Regression equation and median lethal time of *A. spiniferus* at −7 °C.

Instar	Regression Equation	Median Lethal Time (95% Confidence Interval)/h	*R* ^2^
1st instar	Y = −4.079 + 2.701X	32.353 (11.134~48.337)	0.9690
2nd instar	Y = −4.642 + 2.914X	39.179 (26.794~51.497)	0.9699
3rd instar	Y = −3.417 + 2.196X	35.971 (29.876~42.308)	0.9721
4th instar	Y = −3.75 + 2.281X	44.054 (37.327~52.402)	0.9682
Pseudopupae	Y = −3.732 + 2.346X	38.973 (32.847~45.573)	0.9633
Adult	Y = 3.692 + 3.342X	2.8127 (2.2882~3.3914)	0.9457

## Data Availability

The datasets generated and analyzed during the current study are available from the corresponding author on reasonable request.

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
