# Peer review of "Brief Warm and Aldo-Keto Reductase Family AspiAKR1B1 Contribute to Cold Adaptation of Aleurocanthus spiniferus"

_insects, 2025, doi:10.3390/insects16010038_

Round 1

Reviewer 1 Report

Comments and Suggestions for Authors

See attached.

Reviewer 2 Report

Comments and Suggestions for Authors

In this study, authors studied the role of brief warm pulse and AspiAKR1B1 in Aleurocanthus spiniferus cold tolerance, they also measured the contents of some cryoprotectants in A. spiniferus. The study is comprehensive, but it still needs major revisions before acceptance. The writing is not good, the language needs to be carefully revised. Some questions and comments are listed below.

In this ms, two phrases, cold resistance and cold tolerance, were used in different sections. However, resistance is not the same as tolerance. Authors should review relative research articles and revise it in ms.

In Introduction section, the first two paragraphs should be merged into one.

Why did different tests select insects with various stages? For example, SCP and FP--3rd and 4th instar, and pseudopupae; Low-Temperature Survival Rate--nymphs, pseudopupae and adults; Brief Warm Pulses Test--First and second instar nymphs and overwintering nymphs (3rd and 4th instar); physiological contents--4th instar.

In this study, AspiAKR1A3 also showed a high expressed level in winter, why did the authors not identify the function of the AspiAKR1A3 in insect cold tolerance?

In supplementary, primers of negative control for RNAi should be provided.

Line 14: damages, causes

Line 17: examines, impacts

Line 21: instar

Lines 22-23: in November

Line 23-24: Reformulate this sentence.

Line 24: a comma should be added after “Conversely”.

Line 27: shows a significant decrease

Line 32: Choose nouns or noun phrases as keywords, cold resistance is better.

Line 39: a comma should be added after “grapes”.

Line 41: A. spiniferus

Line 42: Awkward sentence structure. Rewrite.

Line 50: To establish a species? Awkward sentence.

Line 50: factor

Line 57: delete the

Line 61: a comma should be added after winter.

Line 64: “ inositol, and fructose”

Line 64-65: enhancing

Line 65: damages

Line 66: Rewrite this sentence.

Line 79: A. spiniferus

Line 80: effects

Lines 95, 104, 109: individuals is better than insects.

Line 98: Add a comma after pseudopupae, which is named as Oxford Comma.

Line 98: Nymphs, 3rd and 4th instar? Please clarify the instar of insects used in study.

Line 98: Why authors collected insects in July, not in other months?

Line 101: Awkward sentence structure. Rewrite.

Line 105: Where did Brief Warm Pulses Test performed? Environmental incubators or other equipment? Please clarify.

Line 107: Add a comma after this.

Line 118: tested insects

Line 122: Add a comma after hours

Line 133: For “ Polyol and Sugar Content”, why the insects that used in this test was collected in November and March, not other months?

Line 174: tested insects

Lines 158-176: Whats your negative control in RNAi, dslta? Please provide citations. Why not use dsEGFP as a negative control? Another, I can not find the primers of this gene in supplementary.

Line 178: with GraphPad

Lines 187-189: Awkward sentence structure. Rewrite.

Lines 185-194: These sentences are not consistent with the Figure 1. Figure 1 shows the survival rate of specific stage (the 1st, 2nd, 3th, 4th, pseudopupae, and adults) at -7℃ when they were exposed to different times. Please restructure.

Lines 205-210: Based on the order in figure 2, you should present information in figure a and b first, then figure c and d.

Line 219: among the different developmental stages of A. spiniferus natural populations in November.

Line 221: adult A. spiniferus

Line 221: higher. The values of SCP and FP of adult A. spiniferus were higher than those of other stages, which indicates that cold tolerance of adult A. spiniferus was lower than other stages.

Line 224: In November,, add a comma.

Line 227: Similarly,

Line 227: The month should be added.

Line 224-234: lower. The values of SCP and FP were lower in November than in other months.

Line 246: levels of ... or contents of ....

Line 247-248: Awkward sentence structure. Rewrite.

Line 260-264: Awkward sentence. Authors should clearly indicate the picture in figure 6, such as figure 6a, b, e, etc..

Line 265: What is the dslta treatment? Please see previous comments.

Line 267: temperatures

Line 267-269: Awkward sentence structure. Authors have stated the P-value (P=0.009, 30 min; P=0.008, 60 min), why dose (P>0.05) reappear at Line 269?

Line 259: Authors just identified the function of AspiAKR1B1 in cold tolerance, so the subtitle The Function of AspiAKR1... is inappropriate.

Line 285-286: “ -7℃ was identified as the lowest temperature that A. spiniferus could tolerate.” needs citations. In addition, this sentence starts with “-7℃”, which is inappropriate, please revise.

Line 289: these findings ...

Line 293: temperatures, were

Line 295: “ability of A. spiniferus”

Line 298: in November.

Line 301-303: Is there any correlation between this sentence and previous one? This sentence might be more appropriate on Line 315.

Line 307: suggests

Line 308: increased

Line 309: declined

Line 315: adapt to

Line 316: “glycerol content in A. spiniferus nymphs

Line 320: Add a comma after for instance.

Line 320-321: Rewrite this sentence.

Line 323-325: This sentence should be placed on Line 322.

Line 327: Authors should have made it clear that it was insect cold tolerance.

Line 328: Comment is same as Line 327.

Line 331: Add a comma after “Alphitobius diaperinus”

Line 335: increased

Line 335: However,

Line 338: cold-resistant genes

Line 341: functions

Line 343: Add a comma after “(Xenopus laevis)”.

Line 344: . not ,

Line 349: “the survival rate, SCP, and FP”

Line 348-351: Rewrite this sentence. As authors have mentioned at Line 265-267 in ms, when AspiAKR1B1 was knocked down, the super-cooling point (SCP) increased.

Line 352: “catalyzes

Line 353: coincided

Line 354: “aids

Line 364: Delete paper.

Comments on the Quality of English Language

The language needs to be carefully revised, especiall grammar and tenses.

Reviewer 3 Report

Comments and Suggestions for Authors

The paper by Jia et al. presents the biological roles of AspiAKR1B1 that favoring the orange spiny whitefly to cope with cold adaption. 

Comments

1. Lacking some of the key information in method section, such as in Line 155, 10-fold dilution, what is the final concentration of templates. You need to point out what is the reference genes in qPCR test? How to calculate the relative expression level of the genes. Missing the definition of  the control samples. You cannot using the expression level instead of relative expression level.

2. Pay attantion to that: the gene name and Latin name of insects should be italic.

3. The reasons why you guys conducted the experiments with the directly collected insect samples, instead of the experimental rearing insect samples. Prior to experiment, how did you exclude the factors of abiotic and biotic effects on your experimental results, especially in the ecological and the indoor molecular experiments.

4. The format of statistical in results should be corrected.

5. In my opinion it is not suitable to make the compaision of the same stage of insect between different months.

6. I suggested to adding the RNAi experimental evidence of  gene AsprAKR1A3 to exculde its possibility on cold adaption. Becuase this gene is highly upregulated under the cold stress. In comparisions, the relative value of AsprAKR1B1 just slightly upregulated.

Comments on the Quality of English Language

7. The English of the entire manuscript should be improved.

Round 2

Reviewer 2 Report

Comments and Suggestions for Authors

Overall, the authors did well attending to the reviewers comments. I only have minor

additions and edits that I view as necessary revisions.

Line 121: Each

Line 196: The

Line 221: Data in Table 2 should be consistent. Tables should not span pages.

Line 297: knockdown, the first letter should be lowercase.

Author Response

Dear Professor. Thanks for your attention. The review comments entered seems to be incorrectly, as all comments are inconsistent with the manuscript.

Reviewer 3 Report

Comments and Suggestions for Authors

1. Missing simple summary.

2. Abstract: Rewrite L15-18. Please focused on the cold adaption of the pests and clearify the reasons why using the brief warm pulses to improving their cold adaption?

                   Using some of exact values to describe the results.

                   L32  Lower latitudes?

3. Introduction: Add the meaning of this current study in the end of this section.

4. Please carefully respond my comments on the first time review with the reasonable explanation. 

5. Be carefully the English of the entire text. some English errors.  L123 four instar instead with fourth instar.

Comments on the Quality of English Language

Be carefully the English of the entire text. some English errors.  L123 four instar instead with fourth instar.

Round 3

Reviewer 2 Report

Comments and Suggestions for Authors

Authors have revised ms carefully. I have no comments and recommend acceptance.

Reviewer 3 Report

Comments and Suggestions for Authors

Extend the length of simple summary.